# Indirect Determination of Residual Prestressing Force in Post-Tensioned Concrete Beam

**DOI:** 10.3390/ma14061338

**Published:** 2021-03-10

**Authors:** Jakub Kraľovanec, Martin Moravčík, Petra Bujňáková, Jozef Jošt

**Affiliations:** 1Department of Structures and Bridges, Faculty of Civil Engineering, University of Žilina, Univerzitná 8215/1, 01026 Žilina, Slovakia; martin.moravcik@uniza.sk (M.M.); petra.bujnakova@uniza.sk (P.B.); 2Laboratory of Civil Engineering, University of Žilina, Univerzitná 8215/1, 01026 Žilina, Slovakia; jozef.jost@uniza.sk

**Keywords:** prestressing force, saw-cut testing, assessment, in situ test

## Abstract

A diagnostic survey on the precast prestressed bridge Nižná confirmed significant deterioration due to environmental distress. Evidently, decisive failures of the structure have a similar character as in the previous precast prestressed bridge in Podbiel in the northern part of Slovakia. These failures result from the unsuitable concept of the first generation of precast prestressed concrete beams, which was used in the former Czechoslovakia in the second half of the 20th century. Subsequently, experimental verification using the proof-load test was also executed. This bridge was built in 1956, so at the time of testing, it was 60 years old. The paper presents the indirect determination of prestressing level in one precast post-tensioned concrete beam using the saw-cut method. Experimental measurement was executed during the bridge demolition. Subsequently, a 2D numerical model in ATENA 2D Software, with the assumption of nonlinear material behavior for verification of experimental results, was performed. Finally, the residual prestressing force was evaluated and compared with the expected state of prestressing according to Eurocodes after 60 years of service.

## 1. Introduction

In the early 1960s in former Czechoslovakia, the precast and prestressing technology have initiated to develop [1,2]. Currently, these precast post-tensioned bridges are approaching 60 years of service. Their present structural condition is inappropriate, closely related to their maintenance, inadequate inspection, external loads, and careless conceptual design [2,3]. The anchorages were not made of corrosion resistant steel and they were insufficiently embedded and concreted [4]. Due to the detected failures, three of these bridges on the important international road No. I/59 to Poland were structurally deficient and immediately closed and demolished. A typical representative of the first generation is a T-shape precast prestressed concrete girder with longitudinal and transversal prestressing, creating an orthotropic system. Transversal prestressing was normally conducted in the diaphragms and upper flanges of precast girders [1,2,3,4]. In this structure, the absence of conventional reinforcement influences the ductility and may cause a brittle failure without any warning [5,6]. The general condition of existing bridges not only reflects the level of development of the society in which they were built, but also the cultural and economic power of the present generation, as they reflect the care for these inherited engineering works [7]. Throughout their life, they require regular maintenance actions, the costs of which are generally supported by the operators and owners [8]. In the case of bridges on road networks, the SRA (Slovak Road Administration) statistics database and available sources show that about 34.2% of existing bridges do not satisfy the current criteria for reliability and 2.5% are in very poor/breakdown condition [9]. This is an urgent problem because the ageing transport infrastructure in most European countries and the United States of America is more than 50 to 60 years old and requires rehabilitation based on objective evaluations [10,11].

Nowadays, the research activities of the Department of Structures and Bridges are focused on reliability assessments and investigation of existing steel and concrete bridges, along with their possible rehabilitation [12]. Corrosion losses have a direct impact on the load-carrying capacity of structural elements and consequently on the overall safety of the bridge structure [13]. The type of coating has an impact on the degradation of corrosion protection [14].

The currently available methods for determining the state of the prestressing force can be classified according to their influence on the structure or approach to the evaluation of the level of prestressing. The classification of some methods is questionable because the line between destructive, semi-destructive, and non-destructive approaches is narrow. In general, methods with no or only minimal impact on the structural integrity are considered non-destructive [15,16,17,18]. Two main stress releasing methods are available. First, the drilling method (also called the stress-relief coring technique) is based on observing strain change in the area adjacent to a relatively small hole drilled in the concrete member (usually concrete box girders or slabs). Total stress or strain relief can be theoretically observed at the edge of the hole, where the radial stress equals zero. However, the strain relief disappears as the distance from the drilled hole increases [19]. Second, the stress release method is a saw-cut method, which is key in the presented paper. Its principle is practically the same as in the case of the drilling method. The only difference is that stress relief is initiated by sawing, which either fully or partially isolates the concrete block from the acting forces. Subsequently, stress relief is monitored in the area adjacent to performed saw-cuts. Generally, the concrete block can be considered to be fully isolated if the increasing depth of the performed saw-cuts no longer initiates any major change of the measured strain. Eventually, recorded strain change enables one to determine the residual value of the prestressing force in the investigated structure. If it is possible to determine the ratio of the released strain (stress), for example, by using validated results of a numerical analysis or parametric study, full isolation of the concrete block is not necessary. This is especially true in cases of older bridges with a considerably low concrete covers, which limits the maximal depth of the performed saw-cuts. The flat-jack test can also be applied to massive concrete structures, such as dams. It is possible to assess the local stress state on the surface and simultaneously identify the modulus of elasticity. In the analysis in reference [20,21], the authors used two T-shaped orthogonal slots. All of the abovementioned methods only have a negligible impact on the integrity of the concrete structure and cause local damage that can be properly repaired. Hence, these techniques are generally considered as non-destructive indirect methods for determining the state of prestressing [17,18]. From an extensive failure investigation of the first generation of precast bridges in Slovakia, a precast post-tensioned bridge in Nižná was selected and is discussed in this paper.

## 2. Bridge Survey

Fortunately, early intervention and the decision to close the bridge resulted in the avoidance of fatal consequences. A low level of maintenance quality in combination with heavy traffic on the bridge contributed to the failure of the bridge, in addition to degradation. Due to the failed condition, the Nižná bridge was demolished and replaced with a new one. Before demolition, in situ measurements were taken on the selected beam.

The bridge structure consisted of four simply supported spans with an effective length of 17.18 m; the total length of beams was 17.88 m, see Figure 1. Longitudinal unfilled gaps between the beams were approximately 40 mm wide. The width of the structure was 9.67 m. Transversally, a total of ten beams with a spacing of 0.97 m and a height of 1.05 m were designed. After demolition, only four rebars of conventional reinforcement with a diameter of 6.0 mm were detected, along with stirrups of the same diameter and a spacing of 0.25 m. At the time of construction of the bridge, the rebar type R (Roxor) with a mean yield strength f_y_ of 380.0 MPa was used as a conventional reinforcement.

The precast post-tensioned concrete beam in a visually good condition was selected. This beam seemed to be the least degraded and had a favorable approach from the underside, which was necessary to perform the experimental measurements. Specifically, it was the edge beam in the first span of the bridge, which was already out of order at that time. The beam under investigation is shown in Figure 2. Contrasted to the beams in the other spans of the bridge, it visually exhibited almost no damage, and therefore, it was reasonable to expect a relatively good condition of the prestressing steel as well. Prior to the experiment, the post-tensioned concrete beam was carefully separated from the others, which had already been demolished at the time of the test, and thus, the transverse connection in the form of prestressing was interrupted. For this reason, the selected beam acted independently as a simply supported beam, embedded on the original supports located on the abutment and pier.

During demolition, the remaining material on the upper flange of the beam could not be completely removed. Therefore, in the analysis, it was necessary to consider the load resulting from the material that remained on the beam. Namely, this consisted of a concrete layer with an average thickness of approximately 50 mm. This layer can be seen in Figure 3.

The longitudinal prestressing of the beam was unbonded, and therefore, transfer was provided only in the anchorage area. Very rarely was the cement grout endoscopically discovered, and in such cases this was only locally at the lower surface of the cable duct; even in these areas the grout fell off the prestressing wires after a gentle tapping with a hammer. Hence, it was reasonable to assume that the bond between the cement grout and the prestressing wires was not ensured. From the photographs provided by the endoscope—which was inserted into the cable ducts—it can be seen that the prestressing wires of the selected beam were in good condition almost without any sign of corrosion. This fact confirmed the primary assumption of the good state of the selected post-tensioned concrete beam, determined by visual assessment. Following the experimental measurements, some prestressing wires were also inspected destructively. Once again, it was possible to visually confirm the good condition of the prestressing wires without corrosion. Prestressing wires after exposure are shown in Figure 4 and Figure 5a.

After exposing the anchorage area, considerable corrosion of anchorages was observable. This corrosion can be credited to insufficient protection of the anchorages by the concrete, and thus, to the easy access of water to this area. The anchorage area is shown in Figure 5b.

## 3. Experimental Program

In cases where the material properties are not available, it is almost impossible to determine the actual state of prestressing [22,23]. Hence, the determination of material properties was part of the experimental program, as this was a necessary basis for numerical analysis and evaluation of residual prestressing. For determining the value of residual prestressing force, an indirect method was applied, specifically, the saw-cut method.

### 3.1. Material Properties

#### 3.1.1. Concrete

Concrete’s compressive strength was roughly determined non-destructively based on Schmidt hammer rebound. The obtained results are listed in Table 1 and Table 2. This method is suitable for in situ primary measurements. Estimation of the concrete strength, based on the reading obtained from the measurement using the Schmidt hammer method, was performed as a part of the diagnostics procedure before demolition. Thus, the procedure was applied on six different beams (including the investigated beam). Finally, the concrete strength class was evaluated from the measurements, in compliance with [24,25]. The obtained and calculated results—based on the Schmidt hammer rebound test—indicated a concrete strength class of C30/37.

β_n_ = 1.98; s_r_ = 2.94
R_bg_ = R_be,i,avg_ − β_n_ × s_r_ = 40.11 MPa → C30/37(1)

Since this method does not provide the exact compressive strength of concrete, three cylinders with a diameter of 100 mm were removed from the beam prior to demolition and later tested in the laboratory. This test was particularly important in order to obtain the secant modulus of elasticity of concrete E_c_, which is necessary for the calculation of stress based on the strain measurement. Nevertheless, in the case of determining the residual value of prestressing force on the structure—which should be put into operation again—it is suitable to use mainly non-destructive methods for determining the mechanical characteristics of materials. In research practice, the most suitable approach is the combination of non-destructive, semi-destructive, and destructive methods and the subsequent comparison of the obtained results.

As a result of concrete mechanical properties testing, the cylindrical compressive strength f_c,cyl_ was 31.5 MPa and the secant modulus of elasticity E_c_ was 33,980 MPa. The results are presented in Table 3. Ultimately, for this experiment, the values of the compressive and tensile strength of concrete are not strictly required, whereas they should not affect the local stress change in the monitored area.

The content of chloride ions above a certain limit significantly increases the risk of corrosion of the reinforcement. Therefore, in situ measurement of the chloride ion content of concrete was provided using the rapid chloride test (RCT) system by Germann Instruments [26]. This method is used to determine the amount of acid-soluble chlorides in powder samples of concrete. Four powder samples were mixed with extraction liquid, which removes disturbing ions (e.g., sulfide ions) and extracts the chloride ions. The measurement is based on millivolt reading, and then, the value is plotted on the calibration curve. Subsequently, the chloride content of concrete was read from the curve (Cl^−^/m_s_) and recalculated to the estimated quantity of cement (Cl^−^/m_c_) [26]. Obtained values are listed in Table 4. All samples contained a higher chloride ion content than the critical value according to Eurocode 2 (0.40 Cl^−^/m_c_) [27].

#### 3.1.2. Prestressing Wires

After demolition of the beam, the anchorages and prestressing wires were removed for material testing. Subsequently, based on the laboratory tensile test, the mean measured tensile strength of the prestressing wires f_pt_ was 1647 MPa, the mean yield strength f_y_ was 1160 MPa, and the modulus of elasticity E_p_ was 190,000 MPa. The cross-sectional area of a wire was 15.9 mm^2^.

### 3.2. Evaluation of Residual Prestressing Force

The saw-cut method was applied to determine the residual prestressing force. The scheme of a selected post-tensioned precast concrete beam is illustrated in Figure 6 and Figure 7. Thirteen prestressing tendons, consisting of nine patented wires with a diameter of 4.5 mm, were placed in a cable duct with a diameter of 30 mm, see Figure 7c. Cable ducts were identified as metal tubes.

Two pairs of saw-cuts with an identical axial distance of 120 mm were performed in the beam, see Figure 8. The pair of saw-cuts SC1 had a depth of 23 mm and the pair of saw-cuts SC2 were 31 mm deep. The applied saw-cuts are shown in Figure 9. These parameters were determined by the measurement of the saw-cuts’ depth and axial distance after the actual sawing of the beam and were the necessary basis for further analysis. Our intention was to avoid cutting the beam’s reinforcement. Linear foil strain gauges HBM LY41-50/120 made of ferritic steel (temperature matching code “1”: 10.8 × 10^−6^/K) with a measuring grid length of 50.0 mm and a total length of 63.6 mm were used to record the strain. The linear foil strain gauges were installed in the middle of the axial distance between the saw-cuts. Transversely, they were applied to the center of the lower flange of the beam. The measurement was finalized after stabilization of the recorded values on both used linear foil strain gauges. The maximum value of the change in stress Δσ, measured by the strain gauge SG1 (related to the pair of saw-cuts SC1), was 3.13 MPa and the strain gauge SG2 (related to the pair of saw-cuts SC2) was 4.21 MPa.

## 4. Numerical Analysis

As part of the evaluation of the residual prestressing force, a nonlinear numerical analysis in ATENA 2D Software (version ATENA 5.7.0n, Červenka Consulting, Prague, Czech Republic) was performed [28,29,30]. In our conducted research, we compared the results from 2D and 3D models for saw-cuts applied along the entire width of the prestressed concrete beam (the length is significantly larger than the dimensions of the cross-section) and found that in this case of stress relief investigation, it is sufficient to use a 2D model. The local stress relief is observed only in the bottom flange of the beam. Consequently, this method of modelling appears to be sufficiently accurate in these types of experiments. However, in the case of prestressed concrete slabs or box girder bridges, it is necessary to analyze stress relief in a 3D model. The same is true for the evaluation of stress relief when using, for example, the drilling method.

The post-tensioned precast prestressed concrete beam was modelled as a 2D finite element, which is shown in Figure 10, while the individual parts of the beam (macro-elements) were assigned the dimensions according to the equivalent cross-section presented in Figure 11. The equivalent cross-section has the same area and second moment of inertia as the real cross-section. The load from the unremoved material with an average thickness of 50 mm on the upper flange of the beam was considered by a uniformly distributed load with a value of 1.15 kN/m. The supports were modelled at a distance of 350 mm from the ends of the beam, and thus, the effective length of the beam in question was 17,180 mm. Based on the results of the bridge survey, the prestressing tendons were considered unbonded. In the middle area of the beam, at a width of 765 mm (the area adjacent to the saw-cuts), the mesh was smoothed into quadrilateral CCQ10SBeta elements with a size of 10 mm. The rest of the modelled beam was composed of quadrilateral CCQ10SBeta elements with a uniform size of 100 mm. The application of the saw-cuts was modelled in the 2D numerical analysis using “construction stages”. First, all macro-elements had the properties of the beam’s concrete. In the next phase, the modulus of elasticity of the macro-elements that represented the saw-cuts was rapidly reduced, thus taking into account the sawing.

The following material parameters were considered in the 2D numerical analysis:Concrete (SBeta Material):
f_c,cube_ = 37.0 MPa; f_c,cyl_ = 31.5 MPa; f_t_ = 2.7 MPa; E_c_ = 33,980 MPa; ν = 0.20
Prestressing Wires (Reinforcement–Bilinear):
f_y_ = 1160 MPa; E_p_ = 190,000 MPa
Saw-cuts (Plane Stress Elastic Isotropic):
E_SC_ = 1.0 kPa; ν = 0.30

The stress–strain curves that were used are presented in Figure 12. The SBeta constitutive model of concrete includes 20 material parameters. Whereas simulations of real behavior were performed, known parameters were chosen based on material testing. Other parameters were based on guidelines for Finite Element (FE) analysis of concrete structures in ATENA Software. The formulation of constitutive relations is considered in the plane stress state. The concept of the material model SBeta includes, for example: the non-linear behavior in compression (including hardening and softening); the fracture of concrete under tension, based on the non-linear fracture mechanics biaxial strength failure criterion; a reduction in compressive strength after cracking; the tension stiffening effect; a reduction in the shear stiffness after cracking; and two crack models—the fixed crack direction and rotated crack direction. More information can be found in [28,29,30].

The principle of the numerical analysis was based on an iterative change of the value of prestressing force until a satisfactory agreement with the experimentally measured results was achieved. The results from the strain gauge SG1 corresponded with a numerical analysis of 99%. This value was 96% in the case of the strain gauge SG2 (Table 5). The stress values were calculated at eleven points over a length of 50 mm, which was chosen to express the length of the strain gauge used during the experimental measurements. The value of the change in stress was calculated as the average of the values obtained at the individual monitoring points.

The values of stress before (σ_c,0_) and after the application of saw-cuts on the beam (σ_c,1_), and the difference between these two values (Δσ_c_) are presented in Table 5. The numerical analysis showed that the used parameters of the saw-cuts partially isolated the concrete block from the acting forces. In the case of the pair of saw-cuts SC1, the stress change was less than 63%, while the pair of saw-cuts SC2 initiated a stress relief of almost 87%. For this reason, the initial stress value in the bottom of the precast post-tensioned concrete beam was determined analytically. Subsequently, the ideal cross-sectional characteristics were calculated and the value of the residual prestressing force in the investigated beam (P_m60,actual_), as a result, was determined using Equation (2). The ideal cross-sectional characteristics of the beam (see Figure 13) are listed in Table 6.

The stress before and after application of saw-cuts is shown in Figure 14. Based on the numerical analysis, the initial stress value in the monitored area (σ_c,0_) was –5.04 MPa.

M_G_ = 327.45 kNm
P_m60,actual_ = −[σ_c,0_ − (M_G_/I_yi_) × z_bi_]/[(1/A_i_) + (e_pi_/I_yi_) × z_bi_] = 945.03 kN(2)

The stress in the tendons, including the expected value of prestressing losses after 60 years of service (21,900 days), was determined according to Eurocode 2 [27]. The value of stress in prestressing tendons during tensioning of 1044 MPa (according to Equation (3)), the end of the curing after seven days, and a relative humidity of 65% were considered in the calculation. The anchorage set was taken into account with a value of 6 mm, which was a common value of the anchorage set at the time of construction of the investigated bridge.
σ_p,max_ = min [0.80 × f_pk_; 0.90 × f_p,0_._1k_] = 1044 MPa(3)

## 5. Discussion

The expected value of the prestressing force based on the expected stress in individual tendons after 60 years of operation (P_m60,theoretical_) in the precast post-tensioned prestressed concrete beam—determined according to Eurocode 2 [27]—was 1090.30 kN. Comparing the value obtained by the standard procedure (P_m60,theoretical_) in compliance with Eurocode 2 [27] and value of residual prestressing force obtained based on experimental measurements (P_m60,actual_) and subsequent nonlinear numerical analysis in ATENA 2D Software, there is a difference of 13.32% (ΔP). The values of the change in stress after application of the saw-cuts, which were recorded during the measurement, are presented in Figure 15. This quite low value of additional prestressing losses after 60 years proves the good technical state of the selected and subsequently analyzed post-tensioned prestressed concrete beam. Endoscopically and later destructively examined prestressing wires showed no notable corrosion, so the prestressing losses can be mainly attributed to the relatively poor state of prestressing anchorages in the ends of the investigated beam. Considering the results, future experiments should be performed using modified parameters of saw-cuts regarding validation of partial and full isolation of concrete block from acting forces.

The expected stress in the individual tendons of the investigated bridge in compliance with Eurocode 2 [27] after 60 years of service is presented in Figure 16. The assumption of the relationship between the value of prestressing force and time is shown in Figure 17.

## 6. Conclusions

This research study focused on the application of the saw-cut method on a precast post-tensioned beam. Based on the in situ performed test and numerical analysis, the following conclusions can be summarized:

One of the advantages of the presented method is that it can be performed with or without the application of an external load. Consequently, if the investigated structure is not loaded by an external load, the determination of the dead load is not complicated. The measurement can be performed quickly and easily in situ and does not require expensive tools.

In the case of older prestressed concrete bridges in service, concrete cover can be immensely low in comparison with today’s standards. This fact can also influence the maximal depth of saw-cuts. Specifically, deeper saw-cuts should initiate full isolation with respect to the structure’s reinforcement especially in the case of large diameter reinforcement. Obtained results suggest that a relatively small intervention into the concrete structure could lead to notable local stress relief. The saw-cut with a depth of 31 mm is negligible in comparison to the cross-section of the beam. Therefore, the integrity of the structure was preserved, and simultaneously, important information including the state of residual prestressing force was recorded. The results indicate that only the 31 mm deep saw-cuts at the axial distance of 120 mm can cause almost full isolation of the concrete block from the acting forces. The saw-cuts with a depth of 23 mm in the axial distance of 120 mm released more than half of the initial stress in the adjacent area. The value of the residual prestressing force, based on the standard evaluation according to Eurocode 2 [27] and the 2D finite element analysis, shows a difference of 13.32%. In practice, the underestimation of the value of prestressing losses with inadequate maintenance with corrosion can lead to unexpected failure of the bridge structure. Consideration of the importance of the real mechanical properties and the average value of stress change at the individual monitoring points represents a key parameter to be taken into account in the numerical analysis.

Given that our findings are based on a limited number of performed saw-cuts, the results from such an experiment should, therefore, be treated with considerable caution. In the future, it is necessary to perform further experimental measurements, both in the laboratory and in situ, with various specimen dimensions and mechanical properties. A large number of ageing, prestressed, short- and medium-span concrete bridges consist of individual beams. Therefore, the future application of this non-destructive approach is very promising.

## Figures and Tables

**Figure 1 materials-14-01338-f001:**
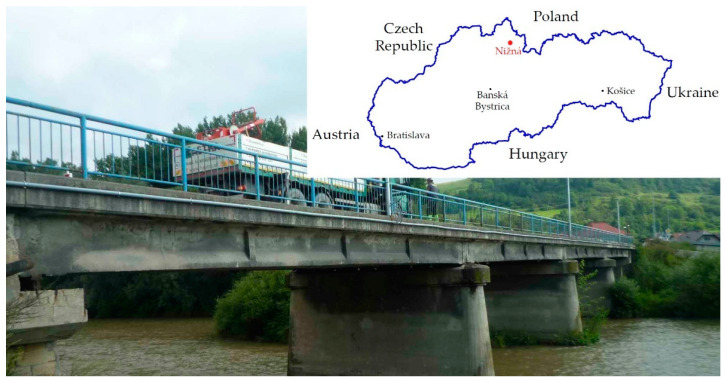
View of Bridge Nižná in service.

**Figure 2 materials-14-01338-f002:**
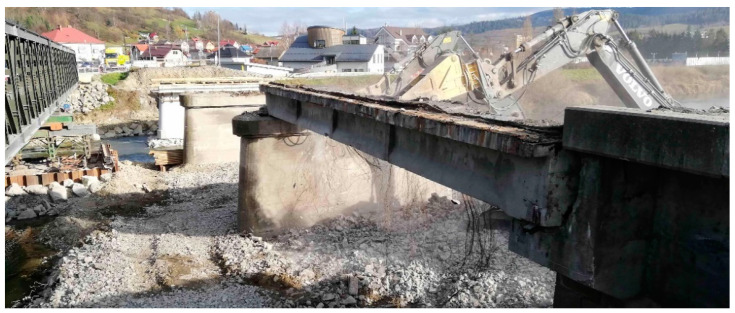
Investigated post-tensioned concrete beam.

**Figure 3 materials-14-01338-f003:**
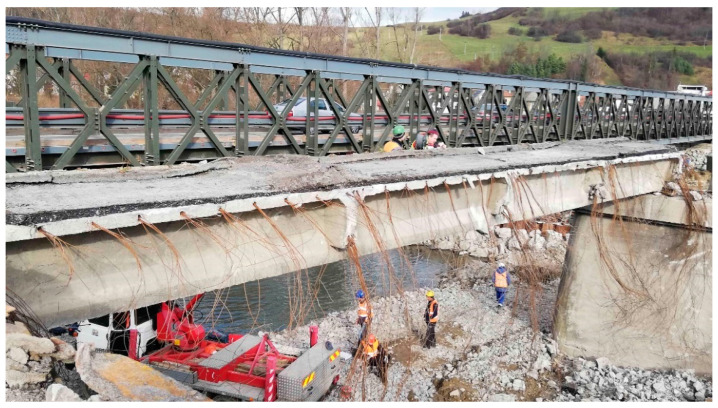
Material on the upper flange of the investigated post-tensioned concrete beam.

**Figure 4 materials-14-01338-f004:**
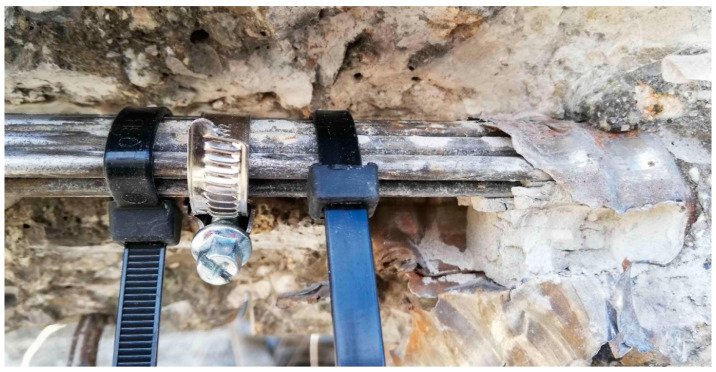
Prestressing wires after exposure (no signs of corrosion).

**Figure 5 materials-14-01338-f005:**
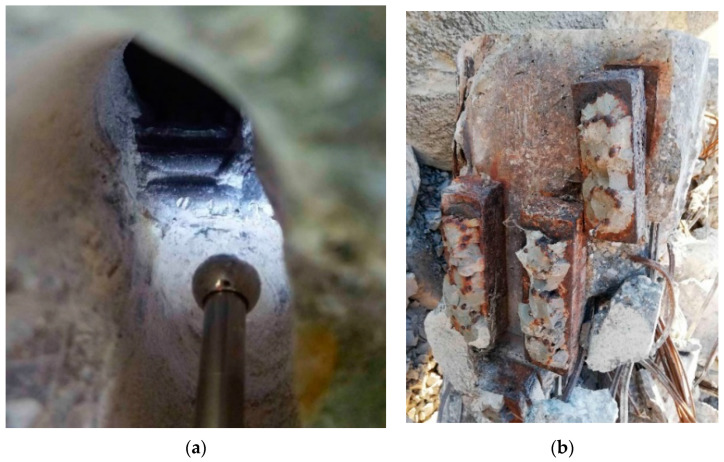
Endoscopic method (**a**); corroded anchorage area after demolition (**b**).

**Figure 6 materials-14-01338-f006:**
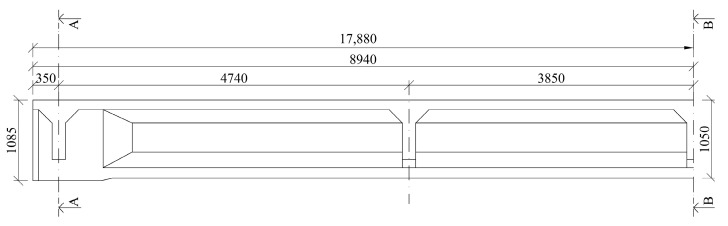
Longitudinal layout of the precast post-tensioned beam.

**Figure 7 materials-14-01338-f007:**
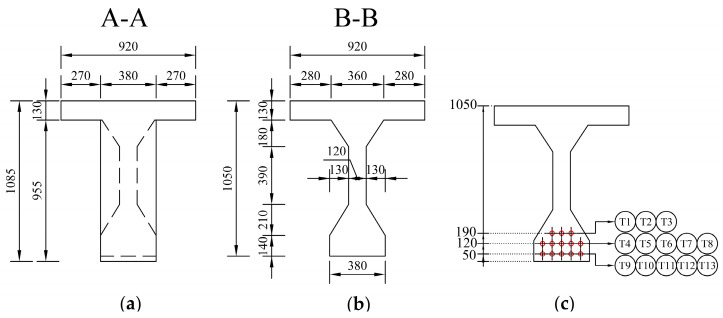
Cross-section of the selected beam: (**a**) at the end; (**b**) mid-span; (**c**) arrangement of tendons.

**Figure 8 materials-14-01338-f008:**
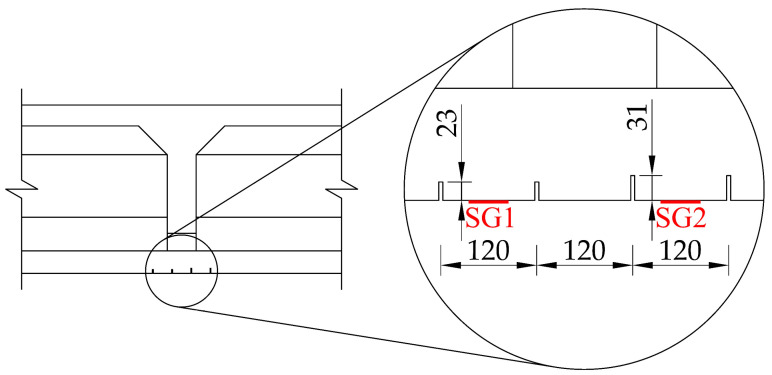
Parameters of performed saw-cuts.

**Figure 9 materials-14-01338-f009:**
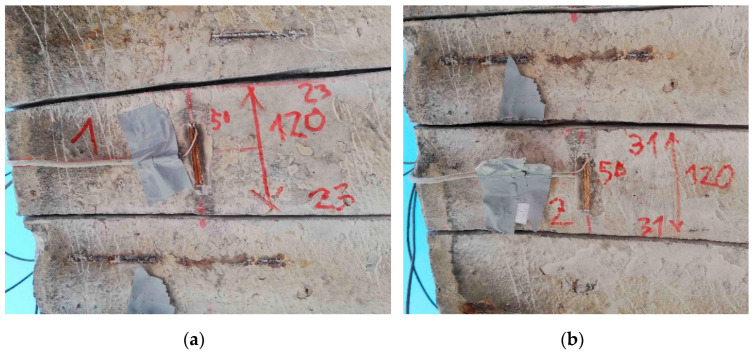
Applied saw-cuts (SC1—(**a**); SC2—(**b**)).

**Figure 10 materials-14-01338-f010:**
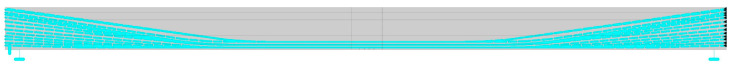
2D finite element model.

**Figure 11 materials-14-01338-f011:**
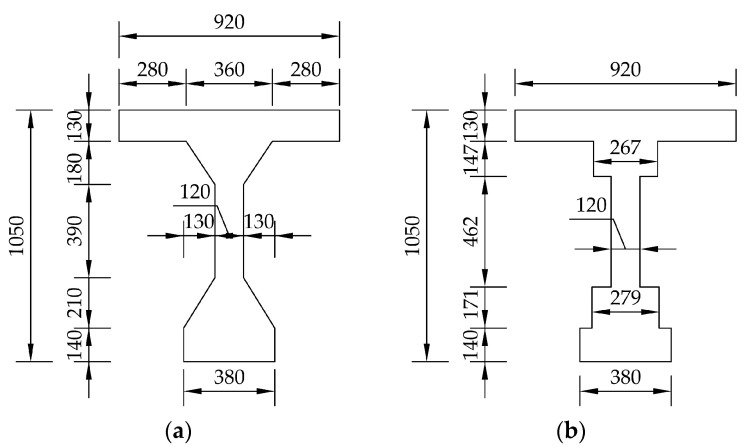
Real cross-section (**a**); equivalent cross-section used in numerical analysis (**b**).

**Figure 12 materials-14-01338-f012:**
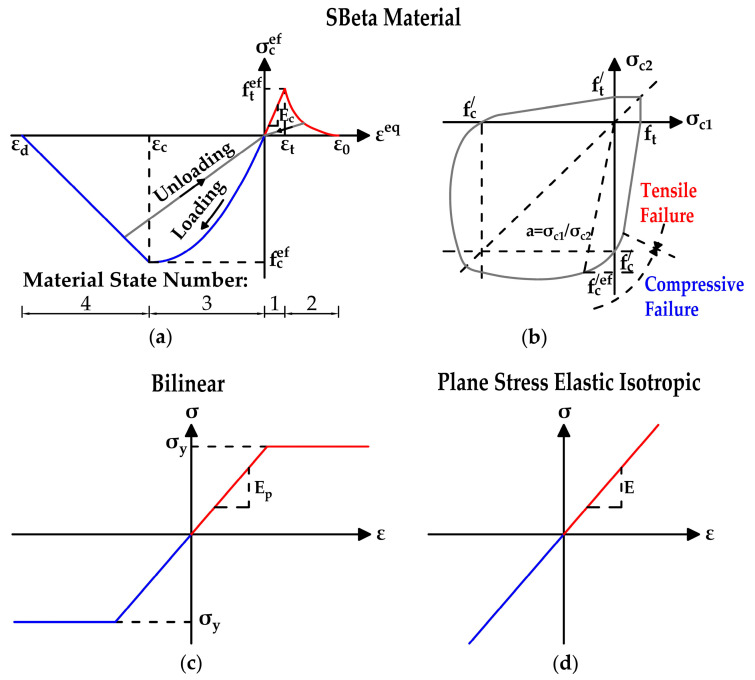
(**a**) Stress–strain curve for concrete; (**b**) Biaxial failure function for concrete; (**c**) Stress–strain curve for prestressing steel; (**d**) Stress–strain curve for saw-cuts [28].

**Figure 13 materials-14-01338-f013:**
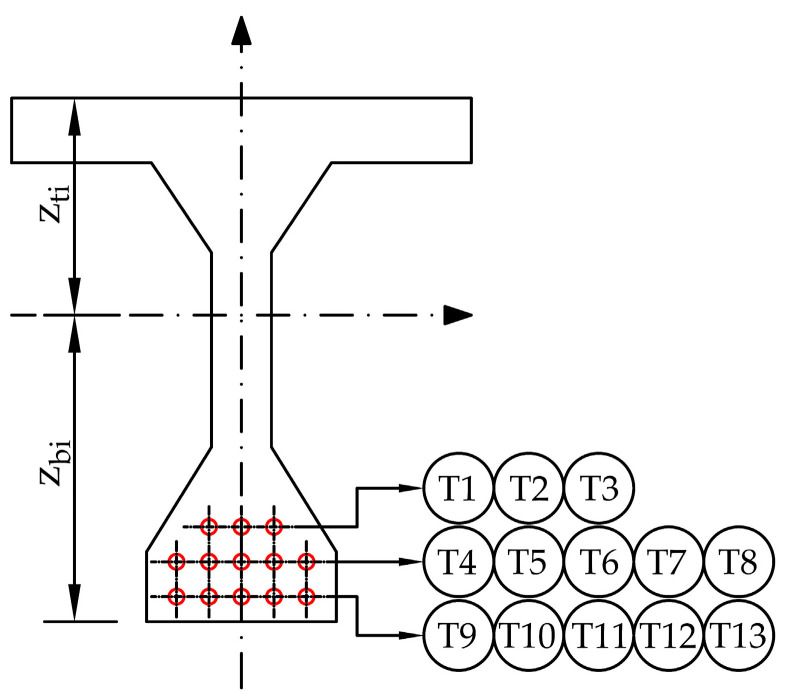
Ideal cross-section of the investigated beam.

**Figure 14 materials-14-01338-f014:**
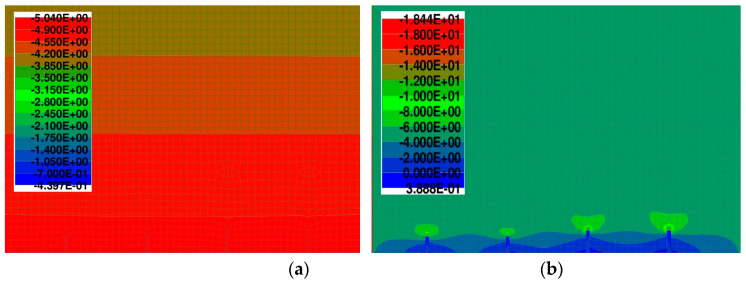
Stress before (**a**) and after (**b**) application of saw-cuts.

**Figure 15 materials-14-01338-f015:**
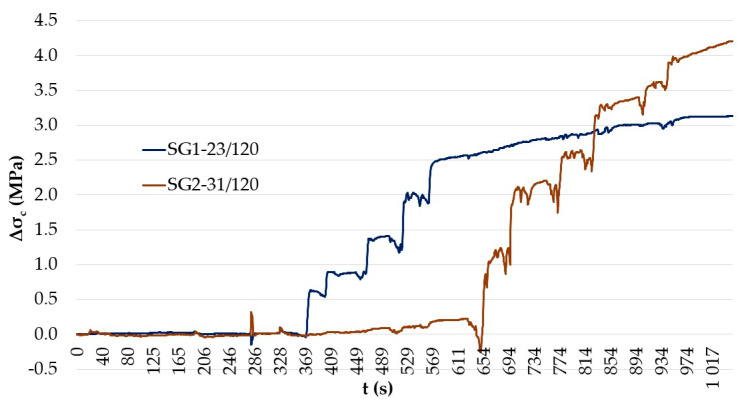
Recorded stress in the bottom edge of the post-tensioned concrete beam.

**Figure 16 materials-14-01338-f016:**
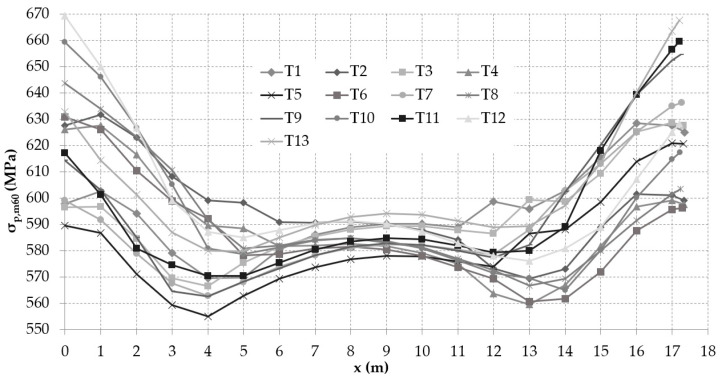
Stress in prestressing tendons after 60 years of service according to Eurocode 2 [27].

**Figure 17 materials-14-01338-f017:**
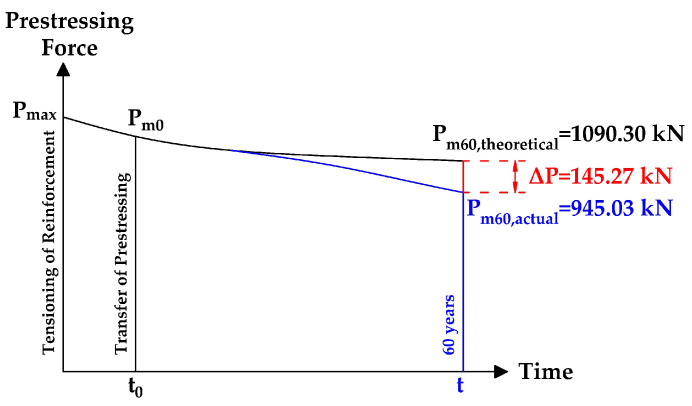
Relation between the value of prestressing force and time.

**Table 1 materials-14-01338-t001:** Schmidt hammer rebounds.

Span/No. of Beam	Schmidt Hammer Rebound a (-)					a_avg_. (-)	R_ci_ (MPa)					R_ci,avg_ (MPa)	α_t_ (-)	α_w_ (-)	R_be,i_ (MPa)
S1/B1 ^1^	45	46	46	50	46	46.6	50	52	52	59	52	53.0	0.9	1.0	47.70
S1/B2	44	49	47	50	50	48.0	48	57	53	59	59	55.2	0.9	1.0	49.68
S2/B9	49	49	49	42	42	46.2	57	57	57	44	44	51.8	0.9	1.0	46.62
S2/B10	46	43	45	42	45	44.2	52	46	50	44	50	48.4	0.9	1.0	43.56
S3/B9	43	43	48	42	47	44.6	46	46	55	44	53	48.8	0.9	1.0	43.92
S3/B10	48	45	44	44	42	44.6	55	50	48	48	44	49.0	0.9	1.0	44.10

^1^ Beam in question. R_be,i,avg_ = 45.93 MPa.

**Table 2 materials-14-01338-t002:** Schmidt hammer rebounds—evaluation according to standards [24,25].

i	R_be,i_ (MPa)	R_be,i,avg_ (MPa)	R_be,i_—R_be,i,avg_ (MPa)	[R_be,i_—R_be,i,avg_]^2^ (MPa^2^)
1	47.70	45.93	1.77	3.13
2	49.68	45.93	3.75	14.06
3	46.62	45.93	0.69	0.48
4	43.56	45.93	−2.37	5.62
5	43.92	45.93	−2.01	4.04
6	44.10	45.93	−1.83	3.35

**Table 3 materials-14-01338-t003:** Mechanical properties of concrete.

Sample	f_c,cyl_ (MPa)	E_c_ (MPa)
CC1	32.6	34,130
CC2	30.9	33,150
CC3	31.1	34,660
**Average**	**31.5**	**33,980**

**Table 4 materials-14-01338-t004:** Chloride Ion Content in samples obtained by drilling.

Sample	Millivolt Reading (mV)	Chloride Ion Content (% Cl^−^/m_s_)	Chloride Ion Content (% Cl^−^/m_c_)
CH1	18.4	0.1980	1.3860
CH2	22.3	0.1862	1.3030
CH3	7.2	0.3010	2.1070
CH4	8.6	0.2950	2.0650

**Table 5 materials-14-01338-t005:** Stress values obtained from numerical analysis.

Saw-Cuts SC1–23/120					Saw-Cuts SC2–31/120				
i	σ_c,0_ (MPa)	σ_c,1_ (MPa)	Δσ_c_ (MPa)	Δσ_c_ (%)	i	σ_c,0_ (MPa)	σ_c,1_ (MPa)	Δσ_c_ (MPa)	Δσ_c_ (%)
1	−5.04	−1.44	3.60	71.41	12	−5.04	−0.33	4.71	93.45
2	−5.04	−1.44	3.60	71.35	13	−5.04	−0.33	4.71	93.48
3	−5.04	−1.99	3.05	60.58	14	−5.04	−0.78	4.26	84.58
4	−5.04	−1.99	3.05	60.58	15	−5.04	−0.78	4.26	84.60
5	−5.04	−2.24	2.80	55.58	16	−5.04	−1.01	4.03	79.96
6	−5.04	−2.24	2.80	55.57	17	−5.04	−1.00	4.05	80.23
7	−5.04	−2.24	2.80	55.51	18	−5.04	−1.00	4.05	80.23
8	−5.04	−2.24	2.80	55.52	19	−5.04	−0.74	4.30	85.35
9	−5.04	−2.00	3.04	60.33	20	−5.04	−0.74	4.30	85.35
10	−5.04	−1.48	3.56	70.62	21	−5.04	−0.27	4.77	94.55
11	−5.04	−1.48	3.56	70.68	22	−5.04	−0.28	4.77	94.52
Avg.	−5.04	−1.89	3.15	62.52	Avg.	−5.04	−0.66	4.38	86.94
Comparison					Comparison				
Δσ_c,EXP_–Δσ_c,NUM_ (MPa)			−0.02		Δσ_c,EXP_–Δσ_c,NUM_ (MPa)			−0.18	
Δσ_c,EXP_/Δσ_c,NUM_ (-)			0.99		Δσ_c,EXP_/Δσ_c,NUM_ (-)			0.96	

**Table 6 materials-14-01338-t006:** Ideal cross-sectional characteristics.

A_i_ (mm^2^)	I_yi_ (mm^4^)	z_bi_ (mm)	z_ti_ (mm)	e_pi_ (mm)
318532.396	4.4532 × 10^10^	614.662	435.338	505.432

## Data Availability

Not applicable.

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
