# Peer review of "Indirect Determination of Residual Prestressing Force in Post-Tensioned Concrete Beam"

_materials, 2021, doi:10.3390/ma14061338_

Round 1

Reviewer 1 Report

The presented manuscript is well structured and well written. The research is design appropriate and the experimental methods are adequately described. However the numerical part is described rather superficially.

1. As the work is mainly experimental, I suggest to extend the numerical part. In my opinion, the use of a 2D model (not explained here if it is a plane stress or plane strain) can hardly be applied to girders with this cross section. An explanation and justification is required.

2. What kid of finite elements were used here?

3. Which formulation?

4. How the prestressed reinforcement was modelled?  

5. line 230-232 "The equivalent cross-section has the same area and second moment of inertia as the real cross-section." - what does it mean?

6. Figure 12 does not explain what specific constitutive model (especially for concrete) was used in the calculations.

7. Conclusions in the form of numbered list are often found in technical reports, but in scientific papers I would recommend avoiding this form.

8. I suggest that the authors present very clearly what scientific value their article brings to the public space, apart from the practical explanation of the techniques used in the expert opinion.

Remark.

I am not sure if the authors are aware of other works involving the smart combination of flat-jack testing and inverse analysis applied to concrete structures, both for parameter characterization and stress reconstruction, e.g.

R. Fedele and G. Maier, Flat-jack tests and inverse analysis for the identification of stress states and elastic properties in concrete dams, Meccanica 42(4), 387–402, 2007

or

T. Garbowski, G. Maier, G. Novati, Diagnosis of concrete dams by flat-jack tests and inverse analyses based on proper orthogonal decomposition, Journal of Mechanics of Materials and Structures, 6 (1-4), 181-202, 2011

Author Response

Dear Reviewer,

thank you very much for your time in preparing the review and comments, which can improve the quality of our article. Our answers can be found in the attachment.

Reviewer 2 Report

The article shows a very interesting and real measurement procedure for a real design solution in combination with numerical modelling.
The saw-cut method in combination with a model in commercial software for indirect preload determination is a very interesting alternative.
I appreciate both the scientific side of the article and the practical applicability of procedures and engineering approaches.
The results can be applied to bridges not only in Slovakia (or aforementioned Czechoslovakia), but in the entire so-called Eastern bloc, where similar structures are located.
I have a few comments on the article:
- The conclusions are clear, but it is not wrong to draw other important conclusions from your work.
- Píšete "it is necessary to perform experimental measurements in the laboratory and in-situ with various specimens' dimensions or mechanical properties". Has this method not been used anywhere yet? Can't the results be compared with other studies?
-Have you tried other software? Is it possible to analyze the problem by procedures independent of commercial programs?

The article can be published after a slight modification.

Author Response

(The authors gave the same response as above.)

Round 2

Reviewer 1 Report

I would like to acknowledge the authors for meeting my comments in a satisfactory way. Therefore, I have no further comments and recommend the manuscript to be accepted for publication in Materials.